# Multi-Frame Labeled Faces Database: Towards Face Super-Resolution from Realistic Video Sequences

**Martin Rajnoha \* , Anzhelika Mezina and Radim Burget \***

Department of Telecommunications, Brno University of Technology, 616 00 Brno, Czech Republic; xmezin00@vutbr.cz

**\*** Correspondence: martin.rajnoha@vutbr.cz (M.R.); burgetrm@feec.vutbr.cz (R.B.)



**Featured Application:** During police searches for perpetrators of serious crimes, often only a low definition video is available. The proposed methodology can use a sequence of faces from a video to reconstruct a single high definition face. Biometric identifications are used worldwide, but there is still space for improvements in their accuracy or working under bad environmental conditions.

**Abstract:** Forensically trained facial reviewers are still considered as one of the most accurate approaches for person identification from video records. The human brain can utilize information, not just from a single image, but also from a sequence of images (i.e., videos), and even in the case of low-quality records or a long distance from a camera, it can accurately identify a given person. Unfortunately, in many cases, a single still image is needed. An example of such a case is a police search that is about to be announced in newspapers. This paper introduces a face database obtained from real environment counting in 17,426 sequences of images. The dataset includes persons of various races and ages and also different environments, different lighting conditions or camera device types. This paper also introduces a new multi-frame face super-resolution method and compares this method with the state-of-the-art single-frame and multi-frame super-resolution methods. We prove that the proposed method increases the quality of face images, even in cases of low-resolution low-quality input images, and provides better results than single-frame approaches that are still considered the best in this area. Quality of face images was evaluated using several objective mathematical methods, and also subjective ones, by several volunteers. The source code and the dataset were released and the experiment is fully reproducible.

**Keywords:** face recognition; super resolution; multi frame; image processing; database; dataset; sequences; deep learning

---

## 1. Introduction

Closed-circuit television (CCTV) is a widely used technology for monitoring public and private places, which helps to increase overall safety, prevent frauds, shoplifting, thefts, burglaries, vandalism, terrorism and others. It is often installed in public areas and businesses throughout the world to prevent the abovementioned crimes and to increase overall public safety. Unfortunately, there are also many abuses and other related questions mainly concerning privacy issues. In relation to police searches and the investigation of serious crimes, there is no doubt that they have a very positive impact on the clarification of criminal offences and early detention of crime offenders [1].

In practice, it often happens that even though the crime is recorded using CCTV, the recording cannot be used for police investigation or a police search and cannot be used as evidence. The cause of this is often the poor quality of the recording, the distance of the camera from the crime place, bad lighting, bad weather conditions, or others factors which cause the person on the recording to be



unambiguously identified. Such a record cannot even be used by the police to ask the public for help in assistance with tracing the perpetrator, for instance in newspapers or online media.

Most of the computer face recognition systems usually use a single still image and based on it recognize a face. The human experts—forensically trained facial reviewers—are still considered one of the most accurate approaches for person identification. They are generally more resilient to noise in images, and also more efficient in utilizing the information contained in the video, even when the quality of images are poor. They can take into account the dynamics of walk, gestures, fitness status of a person, and possibly other characteristics. They can also utilize the information from a sequence of face images and can be more accurate when compared to recognition from each isolated image.

This paper focuses on increasing the quality of the facial image using a video sequence with emphasis on biometric utilization where the principle is illustrated (see Figure 1). The input face image is recorded from a long distance (i.e., its resolution is low and thus a single facial image cannot be used for person identification). This particular example demonstrates the difficulty of person identification from a long distance, which is quite frequent for surveillance cameras. Inspired by human skills, this paper investigates methods that can reconstruct a high-resolution face image from a series of low-resolution and poor quality images.

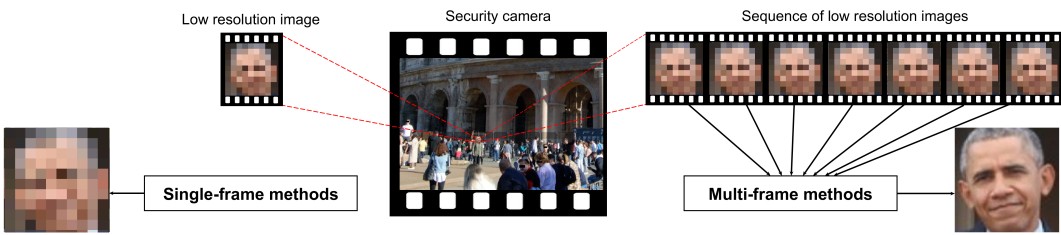

**Figure 1.** This paper is inspired by the human capacity to recognize a face from a sequence of images better than from a still picture.

The research described in this paper introduces a couple of contributions, whose novelty or improvement are briefly described by the following points for better clarity:

- This paper introduces a new methodology for super-resolution (SR) different from the general-purpose one, which is limited to faces and biometric applications.
- The sequence of multiple images is processed (multi-frame methods) instead of using just a single image as most existing datasets allow (single-frame methods).
- We proposed a methodology that was compared to other state-of-the-art methods, currently, the single-frame super-resolution methods can be considered as one of the best because of the lack of the progress and non-utilization of deep learning in multi-frame super-resolution methods [2,3]. Moreover, general super-resolution methods are often not applicable for purposes of biometrics, even if they provide good resolution.
- A new unique large-scale dataset, containing 17,426 samples of image sequences taken from a wide range of recording devices, is provided. They contain faces of various ages, races, illumination conditions that are taken from real-world environments.
- The paper also defines metrics for facial quality measurements that are reproducible and based on open-source solutions. These objective metrics are widely accessible approaches and can boost further research in this area.
- The whole experiment described in this paper is fully reproducible—the source codes and the dataset were released online (http://splab.cz/mlfdb/#download).
- The combination of the key points above creates a ground (basis) and it has the potential of creating a standard for future research in this field.

The rest of the paper is structured as follows. Section 2 provides related work regarding single-frame and multi-frame super-resolution methods. Section 3 describes the experiment which

has been conducted. It includes the description of the created dataset and various proposed methodologies for face recognition and evaluation metrics. Results achieved using our methodology are shown in Section 4, and Section 5 discusses the results we achieved and on how to interpret them. Finally, Section 6 concludes the paper.

## 2. Related Work

Face identity recognition can usually be done manually by forensically trained facial reviewers or automatically, usually by a computer expert system. The automated face identity recognition is a research area which has experienced a rapid growth in terms of accuracy and reliability in recent years. This success is achieved mainly thanks to the utilization of the so-called deep neural networks [4], higher computational power, and also the presence of large training and evaluation datasets such as Labeled Faces in the Wild (LFW) (13,233 images) [5], PubFig (58,797 images) [6], FERET (14,126 images) [7], CelebA (200 k images, 10,177 identities) [8] or Youtube faces DB (1595 identities) [9]. These datasets contain face images with various illumination conditions, expressions, races, and poses. Although there has been great progress in face identity recognition and its accuracy in recent years, the reconstruction of an image from a low-resolution one still remains a challenge [4]. There is currently no large-scale face dataset containing several sequences of images. Many of the current super-resolution methods have a generative nature and, therefore, cannot be used for identification purposes since they are too creative.

The approaches, which addresses the problem of increasing resolution, can be divided by both the content type and input data type. The methods based on the content type can be divided into general super-resolution techniques and biometry-focused super-resolution techniques. From the point of view of the input data type, the methods can be divided into two groups: with a single input image (the so-called single-frame) or with many input images (the so-called multi-frame, i.e., video). Super-resolution, mainly thanks to the success of deep learning, has made significant progress in recent years [10]. This is also the reason why the most successful methods for super-resolution are mainly based on neural networks. In particular, they are Convolutional Neural Networks (CNN) or Generative Adversarial Networks (GANs).

### 2.1. General Single-Frame Super-Resolution Methods

Single-frame super-resolution methods are methods that create a high-resolution image from a single low-resolution image. Most of these methods are general purpose (i.e., they are not specialized to any particular application area, for example, faces).

In case we do not take into account the interpolation methods (such as bicubic, bilinear, nearest neighbour, etc.), one of the first well-known methods was the Super-Resolution Convolutional Neural Network (SRCNN) [11], which was introduced in 2015. This neural network (NN) architecture was relatively lightweight—it had only three layers, which is a relatively low number when considering the current state-of-the-art architectures. However, it was shown that even such a simple NN outperforms significantly the capabilities of any other interpolation method.

Another significant progress in the area of super-resolution was based on the so-called generative adversarial networks. Although GANs were introduced in 2014, they showed their applicability in practice for super-resolution only in 2017. One of the well-known projects was Pix2Pix [12]. It was originally introduced as an image-to-image translation model (the network has the same dimensions of the input and output layers). Especially when compared to interpolation-based methods, Pix2Pix has also shown a significant improvement [12]. SRGAN [13] is another successful approach based on GANs. Its innovation was to use the residual architecture and improvement regarding the loss function—in particular, it used the perceptual loss function during training instead of the pixel-wise loss, such as Mean Square Error (MSE). Another successful architecture was EDSR, which was introduced in [14] and its main idea was to train high-scale models from pre-trained low-scale models. An interesting innovation was to share parameters across different scales. ESRGAN [15] is inspired by

SRGAN. The innovation of ESRGAN is to remove batch normalization layers, the modified perceptual loss function, and transfer learning. The discriminator part of GAN was replaced by a relativistic discriminator [16].

In 2019, mainly due to using huge computational power and large GPU memory, it was shown that the NNs can generate even relatively complex images with a lot of detail [17]. Although these networks have achieved great success, their big disadvantage is the fact that they do not try to faithfully reconstruct the information that is on the input, but they generate only part of the information. For this reason, they are not suitable for biometrics, face reconstruction, and police investigation, as they tend to generate totally new faces. This is unfortunately a problem of most of the GAN based networks.

### 2.2. Single-Frame Super-Resolution Focused on Faces

Super-resolution methods mentioned in the previous section were designed for general use. This section describes a sub-set of super-resolution methods that directly focus on face super-resolution related problems, and in connection with biometrics too. Although many of the methods mentioned earlier offer images in really high-resolution (often $2\times$, $4\times$, or also $8\times$ zoom) and the human eye often cannot recognize that the face was created from the original low-resolution image, these images cannot be used for person identification. The images are synthesized from past experience (images used as a training dataset for NN) and often generates fake faces. The latest comprehensive survey paper related to biometrics and super-resolution techniques is provided in [18]. Among others, this paper also discusses in-depth different requirements regarding general super-resolution methods and biometrics (or face identity recognition) methods.

When we just focus on methods related to facial super-resolution, one of the state-of-the-art methods is the Ultra-Resolution by Discriminative Generative Networks (UR-DGN). This architecture is based on GAN and allows us to upscale images with scale factor 8 [19]. The authors used a deconvolutional network as the generator and a convolutional network as the discriminator. Another approach in this field is the end-to-end trainable Face Super-Resolution Network (FSRNet), which is based on CNN, and FSRGAN, which recovers more realistic textures than FSRNet [20]. These architectures utilize the estimation of landmark heatmaps. One of the latest works is also the Progressive Face Super-Resolution via Attention to Facial Landmark [21]. This method is also based on GAN, however, it uses the progressive method for upscaling the image. Moreover, this work introduces a new facial attention loss, which allows us to restore facial landmarks.

A recently introduced method PULSE [22], which was presented at the CVPR 2020 http://cvpr2020.thecvf.com/), has very encouraging results. Unfortunately, some criticism (for example, Twitter post (https://twitter.com/AlexVasilescu/status/1277009168319143936): "PULSE: Medical and military decision based on hallucinated pixels?") of the method show the disadvantages of the PULSE, and in general already mentioned lack of GANs—its creativeness. There is a demonstration which uses a low-resolution image of Barrack Obama and its SR alternative processed by this method. It created a high-quality face image with no obvious signs that this face was reconstructed by super-resolution. Unfortunately, it reconstructed a totally different person.

### 2.3. Multi-Frame Super-Resolution

Super-resolution has attracted great attention, especially in the area of the entertainment industry the video, or in other words multi-frame. There have been introduced plenty of works which introduce methods based not just on a single input image, but on a sequence of images. These methods have the potential to reduce noise and increase the quality of the images, especially in the area of conversion of old videos into HD resolution where it had great success.

One of the latest works related to general (non-facial related) multi-frame super-resolution is DeepSUM [2]. It is based on CNN, which exploits spatial and temporal correlations. This approach includes image registration inside the CNN architecture. It dynamically computes and applies custom filters to higher-dimensional image representations, instead of compensating for the motion as

a pre-processing step as is common in most multi-frame super-resolution approaches. The dataset from the challenge Proba-V [23] was used, which is dealing with satellite images and allocates important parts using segmentation based maps.

Another method for multi-frame super-resolution is described in [24]. First, all input low-resolution frames are upscaled by the ResNet [25] with scale factor 2. Parallel to that, the low-resolution frames go through the image registration block to determine the sub-pixel shifts. The next step is the application of the shift-and-add fusion to obtain the initial reconstructed image. The final step is the application of the EvoIM process, which consists of iterative filtering of the initial reconstructed image. This method is again general purpose and not facial super-resolution related.

The most frequent utilization of the multi-frame super-resolution is upscaling the video resolution. Frame-recurrent Video Super-resolution [26] is an end-to-end trainable frame-recurrent video super-resolution (FRVSR) framework. It uses the previously estimated high-resolution frame as an input for its following step. Each frame is processed only once, which allows us to reduce the computational cost. The model consists of several steps: flow estimation, upscaling flow, warping previous output, mapping to low-resolution space, super-resolution. It is very important to keep the previously estimated high-resolution frame in the system, otherwise, it is not possible to pass information to future estimations, which can be critical for the reconstruction of the next frames.

TecoGAN [27] is the architecture proposed to solve the following video generation tasks: Video super-resolution (VSR) and Unpaired Video Translation (UVT). The work [27] describes the adversarial learning method for a recurrent training approach, which utilizes spatial contents and temporal relationships. The architecture uses a frame-recurrent generator and a spatio-temporal discriminator. This approach can generate very realistic natural images; however, it can lead to temporally coherent yet sub-optimal details.

Enhanced Deformable Convolutional Networks (EDVR) [28] is a framework proposed in 2019, which allows us to do different image restoration tasks, for example, super-resolution and de-blurring. It consists of the following parts: an alignment module—Pyramid, Cascading and Deformable convolutions (PCD), and a fusion module—Temporal and Spatial Attention (TSA). Moreover, the two-stage strategy is used to boost performance and improve the quality of output frames.

One of the first attempts related to the facial super-resolution method was introduced in 2017 in [29]. The authors proposed an architecture that consists of three modules: feature extractor, face warping, and reconstruction. This architecture restores the central frame of each input sequence utilizing sub-pixel movement and taking into account a number of adjacent frames. This model was tested on the YouTube Faces dataset, which is not dedicated to super-resolution purposes. The authors downsampled all images through $128 \times 128$ px, images were blurred with the Gaussian kernel (2.4), and again downsampled to the size $16 \times 16$ px. The downsampling method is in this case not clear and probably the same method was used for all images, which together with a constant blur, creates a high bias and does not reflect real-world scenarios. For this purpose, there is a risk of over-fitting.

### 2.4. Motivation

Most of the works described earlier are focused on general image super-resolution. Unfortunately, those methods cannot often be used for person identification since facial super-resolution has slightly different requirements.

Although the general super-resolution methods often works very well for general images and sometimes they even provide really outstanding results, these methods still create some mistakes in the images. In the case of general images, these errors can often be ignored. However, a mistake in a face image, where an eye or nose is missing or is deformed, even if it might be from the point of pixel error a minor error, from the point of view of human perception, it can be strongly disruptive. In this case, even a slight modification of the image can have a significant impact on human perception of the face and identification of a person, as shown in Figure 2.

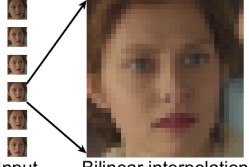 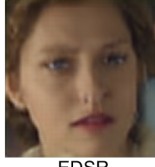 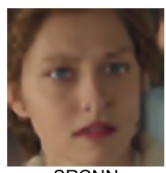 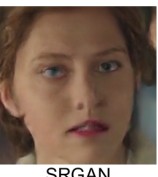 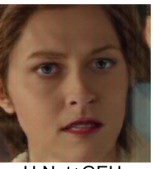

Input　　　Bilinear interpolation　　　　　EDSR　　　　　　　SRCNN　　　　　　SRGAN　　　　　U-Net+GEU

**Figure 2.** Different face super-resolution methods with scale factor 8. From the human perception point of view, it is better to have checkerboard artifacts (U-Net+GEU) instead of artifacts and deformations from other methods.

Another issue may be the credibility of the super-resolution methods from a biometric perspective. For biometrics, it is absolutely unacceptable to be creative and generate parts of the image that do not originate from its input.

The current trend and direction of research of super-resolution methods focus primarily on the general single-frame methods. As the multi-frame methods have more information on its input, it is expected they also probably have a higher potential to reach better results that are based on the input information and are not creatively filling the missing parts of the image. Furthermore, considering the technical capabilities of modern CCTV hardware (e.g., frames independently on fps), it is even possible to go beyond the possibilities of humans and there is an opportunity to reach even better results in the future. Although cognitive skills of the human brain still significantly exceed the skills of machines, narrowly focused artificial intelligence has the potential to bring new applications and, for example, make identification more objective. Examples of these applications could be an automatic search for the best facial image of a suspect for news announcements, reconstruction of the face using a sequence of images, and the suppressing of adverse lighting conditions.

One of the obstacles in the development of the current facial multi-frame super-resolution methods is the absence of a publicly available dataset, which would be designed for this purpose.

Many specific requirements are imposed on such a dataset. Data of such a dataset should not be biased, it must be of sufficient size, it must contain records from real-world environments including different lighting or weather conditions, data must be not dependent on a specific video camera hardware, and it should reflect also real-world compression artifacts, the so-called encoding fragments. Contained faces should cover a variety of human races and various ages. In each case, the data must contain a sequence of several faces, not just one frame.

Regarding available datasets, there is currently no dataset that meets such requirements. Face images used to be biased because of used celebrities, which tends to cause the models to predict "pretty" faces suppressing important individual face features (LFW, CelebA). There is a lack of low quality image inputs, especially in the form of sequences (PubFig, FERET) with a high-quality label (Youtube faces DB). All improvements are summarized in Table 1.

**Table 1.** The improvements of the proposed database in contrast to well-known training datasets.

| Improvement | LFW | FERET | PubFig | CelebA | Youtube DB |
|---|---|---|---|---|---|
| Number of samples | ✓ | ✓ | ✗ | ✗ | ✗ |
| Number of identities | ✗ | ✓ | ✓ | ✗ | ✓ |
| Samples as sequences | ✓ | ✓ | ✓ | ✓ | ✗ |
| Super-resolution purpose | ✓ | ✓ | ✓ | ✓ | ✓ |
| Variability (avoid bias) | ✓ | ✓ | ✗ | ✓ | ✗ |

## 3. Materials and Methods

This section describes the training and testing datasets, examined super-resolution methods, and evaluation metrics. Each particular part is described in detail in the following subsections. Overall, the scheme described in this section and their mutual context are depicted in Figure 3.

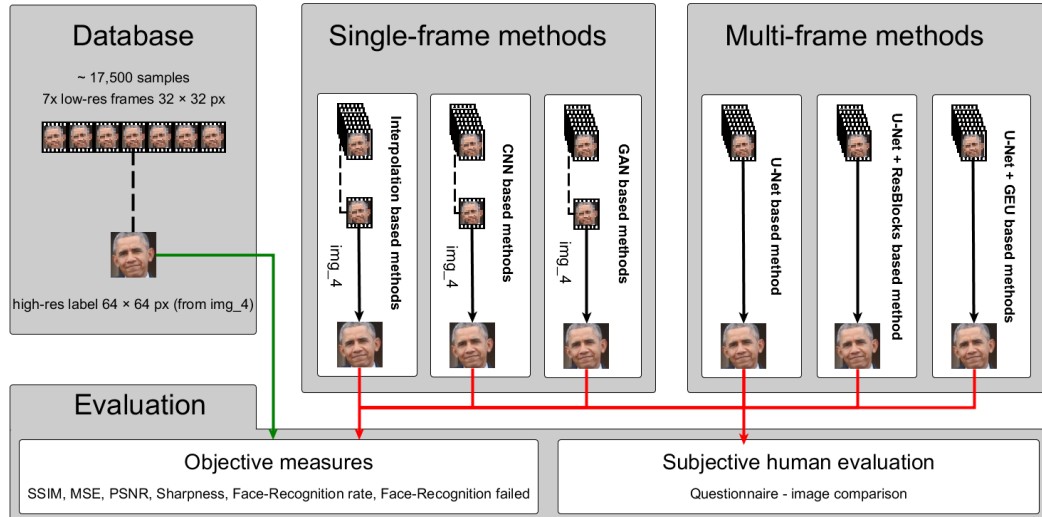

**Figure 3.** Scheme of the experiment. The dataset consisting of several low-resolution images and one high-resolution target image was used for training and evaluation of a couple of different super-resolution methods. These methods were evaluated using several objective and subjective metrics.

### 3.1. Training and Testing Data

For training and evaluation, we introduce a new dataset called Multi-frame Labeled Faces Database—MLFDB. The main difference to the existing datasets is that this dataset does not contain just a single facial image, but it contains a sequence of consecutive frames in a video record. This sequence of images is in a low-resolution ($32 \times 32$ pixels) and serves as an input of the super-resolution methods. Additionally for each of those sequences, there was added one frame in a higher resolution ($64 \times 64$ pixels) which serves as a ground truth, i.e., the optimal output of the super-resolution method (see Figure 4).

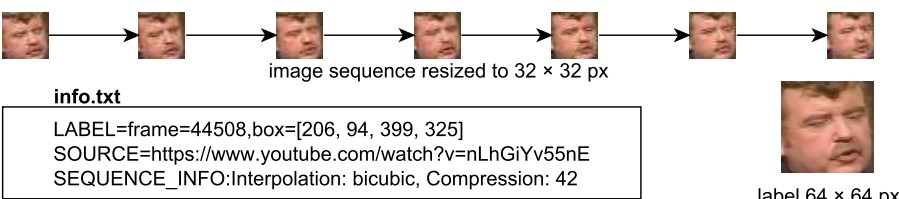

**Figure 4.** An example of sample data. Seven low-resolution images ($32 \times 32$ px) and label ($64 \times 64$ px).

What should be emphasized here is that the faces in the dataset were selected on the borderline where identification is difficult (i.e., each low-resolution frame of $32 \times 32$ pixels) and when it is easier to use it for identification (i.e., higher resolution image with $64 \times 64$ pixels). The main objective is not to have the resulting facial images good looking, but rather that they can be used for identification.

The dataset is of a comparable size, as are the major face datasets (it contains 17,426 faces and approximately 6000–7000 unique persons). It tries to avoid the problem of bias, i.e., the recordings are from various real-world scenarios where faces are nearby to each other, a face is partially covered including glasses, beard, scarf or other coverings (e.g., face paintings in some cases). The dataset also contains various facial images including various lighting conditions (i.e., high, low, and poor lighting quality conditions), various recording hardware was used, videos were encoded using various algorithms, faces contain a wide range of ages, faces were recorded from various poses, and they contain various emotional expressions (including smile, fear, anger or surprise). It also

covers all main ethnic groups including Europeans (i.e., Middle Easterners and Mediterraneans), East Indians, Asians, American Indians, Africans, Melanesians, Micronesians, Polynesians, Australians, and Aborigines.

The created dataset was released online and can be used to reproduce the experiment described in this paper. Although it was primarily designed for multi-frame processing, it is expected its utilization will be wider.

Youtube Faces DB [9] can be considered as a similar dataset to introduced MLFDB. However, it differs in several parameters: Youtube Faces DB contains faces that are not on its borderline when identification is possible and simple downsampling the facial images can bring bias and over-fitting. Moreover, it contains only 1595 different people, but MLFDB contains approximately 6000–7000 different people (see more details in Section 5.1), and also it is not designed for super-resolution purposes.

### 3.1.1. Dataset General Information

The dataset was released online and for the objectivity of the results, a part of the dataset was kept private too. Publicly available sets are TRAIN (12,200 samples) and TEST (2600 samples) all together counting 14,800 face sequences dedicated for training and evaluation purposes. The rest of the dataset (2626 face sequences) is reserved for performance evaluation in private mode. Each sample is organized in an individual folder and in every set, they are numbered from 1—$n$ (TRAIN: 1—12200 , TEST: 1—2600). Each sample folder contains:

- 7 images (i.e., face sequences) in a low-resolution ($32 \times 32$ px) named ,img_1.JPG, . . . , ,'img_7.JPG'.
- The label is defined from the middle of the sequence (from img_4.JPG) with double resolution $64 \times 64$ px and file name label.JPG.
- 'info.txt' file with information about the label (bounding box and frame number), video source, and interpolation method together with the JPEG compression that were used for resizing input images.

A final sample contains: a sequence of seven images, label, and info.txt file (see the example in Figure 4).

### 3.1.2. Dataset Creation

The dataset of sequences of facial images was created with the help of the YOLOv3 object detector (https://github.com/sthanhng/yoloface) [30]. By including only high-quality samples, several filters were applied before the sequence was included in the final dataset. The first filter was the condition that the detected face had to have a resolution greater than $64 \times 64$ px. For reproducibility purposes and also to allow everyone in the future to use the same dataset, all the details from where the faces were captured was provided. This information includes the face in full resolution (used as a ground truth image) together with its bounding box coordinates and video frame number. To assure variability of the dataset only a limited number of faces were collected from each video, and the selected faces were chosen randomly. Next, raw face sequences were extracted from the video using the OpenCV (https://pypi.org/project/opencv-python/) library. It searched for the corresponding face frame (label), it cropped the face according to the given bounding box and finally, it also took 3 frames before and 3 frames after the cropped image using the same bounding box. This was an intentional step on our part to make the dataset more realistic as there are many cases when face detectors do not correctly crop the face (inaccurate detection, crop scaling factor,. . . ). Further optimizations are left for later processing steps. Unfortunately, not all the captured samples can be used. For this reason, a few other steps were performed (see Figure 5):

1.  For each sequence, it is checked whether the label contains a face of sufficient quality so it can be used for face recognition. This was done using an open-source project (Face-recognition

framework [31] v1.3.0, CNN model) which is inspired by Facenet [32]. This step actually validates whether the detected face is in good quality so it can be used for the face recognition similarity rate index metric (see Section 3.4.1). YOLO is a face detector, not a face recognition system, therefore, it also detects such faces where person identification is impossible (e.g., head from behind).

2. In some cases, a sequence can contain multiple overlapping faces. It is not a problem if more faces or their parts occur in one image, but the problem arises, when—mainly due to a mistake of the processing algorithm—there is a different person at the beginning of the sequence and another at the end. From time to time it also happened due to a movie split. Therefore, the label is compared using the Face-recognition framework (used threshold was 0.7) with all other images in the sequence. Thus a possible presence of several faces in a single image was taken into account.

3. Due to used randomness in sampling the videos, it can also happen from time to time that one person appeared more than once in the resulting dataset. The biggest issue has been how to distinguish between very similar images and images with different lighting, face angle, etc. It has been shown that the Face-recognition framework has not been an appropriate solution for this task due to its ability to face alignment, i.e., it can perform the recognition in different conditions. We decided to use a structural similarity metric—SSIM (MSE, PSNR, etc., are not efficient metrics for this case) with the threshold 0.7. Ground truth images are used for this which are first resized into a $64 \times 64$ px resolution.

4. The next step was about validation whether all the sequence images contain a face. This was done using the Face-recognition framework and its method for face detection (not recognition). By using this, only those ground truth images, which can be used for recognition are included. Other images in the sequence must contain a face, but do not have to be recognizable (e.g., a different angle, partially covered by another person, etc.).

5. The sequences which are originally of various resolutions (higher than $64 \times 64$ px) are downsampled into resolution $32 \times 32$ px and the ground truth (label) image into a $64 \times 64$ px resolution. When the original resolution allowed this, the labels were also recorded in resolution $128 \times 128$ px and $256 \times 256$ px). Each sequence randomly used a chosen method for pixel interpolations such as the nearest neighbor, bilinear, bicubic, lanczos (over $8 \times 8$ neighborhood). The label was saved with the best possible JPEG quality (compression 100) and sequence images were saved by a randomly chosen JPEG compression scale from 30–90.

6. Unfortunately, after resizing the ground truth images (labels) into resolution $64 \times 64$ px some of the images were not possible to use for face recognition. Those sequences of face images were removed from the dataset.

# Sequence creation

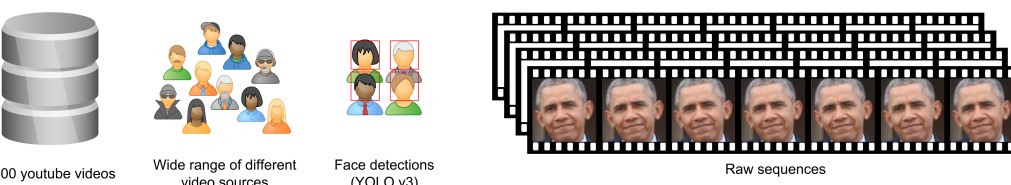

# Database reduction

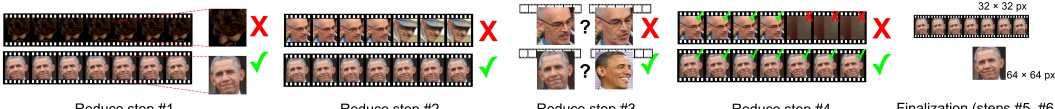

**Figure 5.** Scheme of the process, how the dataset was created. #1 The middle (label) image has to be recognized by Face-recognition, #2 All images in the sequence have to contain the same person, #3 Removed duplicate person records. Same persons with different conditions (light, angle, ...) remain, #4 Every image in the sequence has to contain a face, #5 Sequence and label resizing, interpolation and quality selection, #6 Final validation whether the resulting label can be still recognized by the Face-recognition framework.

## 3.2. Neural Network Architectures

Neural network architectures for super-resolution purposes can be divided into two basic groups: single-frame or multi-frame. The single-frame methods take in our case a single input image of resolution $32 \times 32$ px and reconstructs a higher resolution image with the resolution of $64 \times 64$ px. In other words, it reconstructs an increased resolution of the image with the scale factor 2. The multi-frame methods use several input images, in our particular case 7 images with resolution $32 \times 32$. This sequence of 7 images is used to create a single image of higher resolution $64 \times 64$ (see Figure 3).

### 3.2.1. Single-Frame Methods

For comparison of the proposed methods also with current state-of-the-art single-frame methods, we included a couple of them in this study. They are EDSR, SRCNN, SRGAN, and ESRGAN (all of them mentioned in Section 2.1). These methods were trained and achieved results were compared with proposed multi-frame architectures.

There are also plenty of interpolation methods. These interpolation methods usually underperform when compared to the methods mentioned above. However, since they are frequently used and generally known, they were also included in the study. In particular, they are the bilinear, bicubic and Lanczos interpolation.

All the single-frame methods are designed to have only one input image—in particular, the image number 4 (the middle image in the sequence) since this image is the closest to the label (the label is of a high-resolution).

The single-frame super-resolution methods were evaluated and compared to multi-frame super-resolution methods. This paper aims to demonstrate the potential of multi-frame methods and their utilization in retrieving the information from multiple images instead of using only one image. Since the development of single-frame super-resolution methods has been significant in recent years, especially when compared to the multi-frame methods [3] and also due to the lack of advanced facial multi-frame methods, this paper compares results primarily with the single-frame super-resolution methods [2,3,24,33].

### 3.2.2. Multi-Frame Architectures

All proposed architectures for multi-frame super-resolution have the same basis, which is the U-Net model [34], widely used in the image segmentation area. The architecture of the U-Net model is presented in Figure 6. Since the application of the pure U-Net model for the super-resolution task does not provide satisfactory results, especially for images with faces that require more attention for details, the proposed architectures are designed in such a way that the U-Net model is a part of bigger models. U-Net architecture is one of the models considered in the experiment, which helps to improve the quality of the output image. Moreover, we introduced here some specific modifications of the U-net model. Of course, we experimented with a various number of layers, and also experimented with a couple of subconvolutional layers. Using some blocks was also inspired by previous papers. These modifications and the overall proposed architectures are described below.

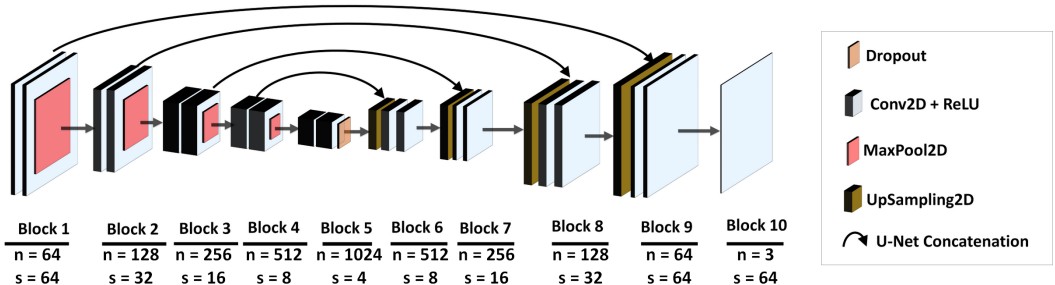

**Figure 6.** The architecture of the U-Net model [34], where *n* denotes number of filters and *s* is features size.

U-Net+Residual blocks architecture: Before input images are fed into the network, they are upscaled using the bicubic interpolation. To extract the features, the combination of convolutional and LeakyReLU layers is used 3 times. After that, the concatenated results go through 5 residual blocks [25] and are added to input image no. 4. The last step is the application of the U-Net network [34]. The difference from the original version of U-Net is the utilization of the Sub-pixel convolutional layer [35] instead of the Upsampling layer in the expanding path. Based on the experiments, sub-pixel convolution can achieve better results in super-resolution tasks and makes the results more accurate. The visualization of the Residual block is presented in Figure 7 and the visualisation of the whole architecture is in Figure 8d. The used loss function is the Mean Square Error (MSE). As it was noticed in other articles [36], this loss function allows us to achieve higher PSNR values, however, the output images have been blurred, because the neural networks with the MSE measure failed to recover details and textures in the high-resolution images [37].

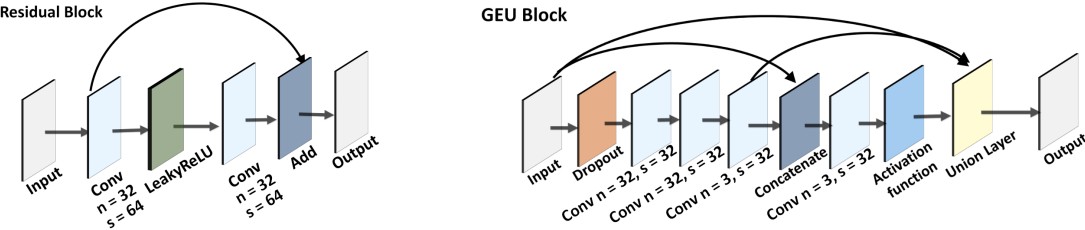

**Figure 7.** The Residual Block [25] and GEU block [38] visualization, where *n* denotes the number of filters and *s* is the features size.

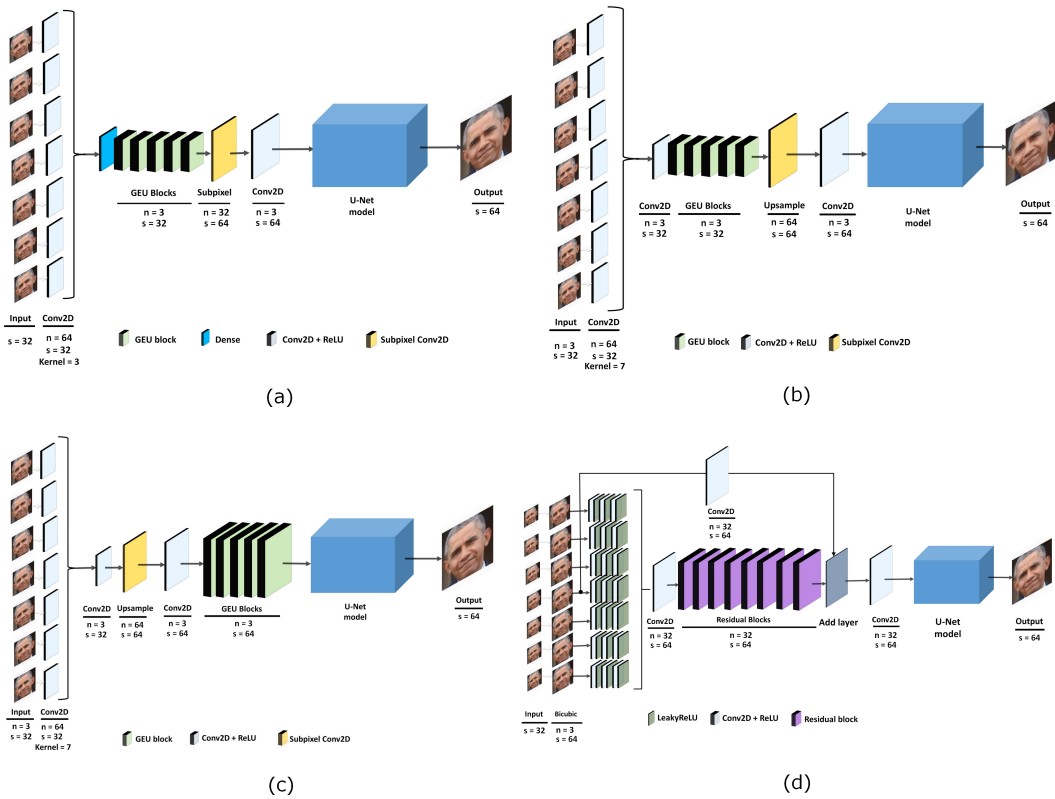

**Figure 8.** Overview of used neural network architectures, (**a**) U-Net+GEU, (**b**) U-Net+GEU2, (**c**) U-Net+GEU3, and (**d**) U-Net+ResBlock, where *n* denotes number of filters and *s* is features size.

U-Net+GEU architecture: This architecture has another proposed method of upscaling the image. In this case, not any pre-processing with an interpolation method was used, but weighted layers were preferred. The so-called Sub-pixel convolutional layer was used during the training of the network. First, the network gets images of low-resolution (32 × 32 px). The result of the concatenation of input layers then goes through individual GEU blocks [38]. The GEU block was inspired by the residual block, which has shown a good performance in previous works [38]. However, the authors of the GEU block modified the residual block in such a way that the input and output are not just added, but they are added using learned weight values—the so-called Union layer:

$$\text{Union layer} = g \times y + (1 - g) \times x, \tag{1}$$

where *x* denotes the input layer, *y* denotes the third convolutional layer and *g* denotes the sigmoid activation layer. The visualization of the GEU block is shown in Figure 7 and the visualization of the whole architecture is shown in Figure 8a.

The output of 5 GEU blocks is upscaled by Sub-pixel convolutional layers. After upscaling, the U-Net model is used with the same modification as in the previous model: utilization of Sub-pixel layers instead of up-sampling. The loss function was modified here which is inspired by the feature reconstruction loss [39], which is shown below:

$$\ell_{feat}^{\theta,j}(\hat{I}, I) = \frac{1}{C_j H_j W_j} ||\phi_j(\hat{I}) - \phi_j(I)||^2, \tag{2}$$

where *I* denotes target image, $\hat{I}$ denotes predicted image, $\phi_j(\hat{I})$ denotes activations of *j*th layer of the network $\phi$ for processing the image *I*. $C_j H_j W_j$ denotes the shape of the feature map. This loss function is from the family of perceptual loss functions, which compute a loss of a pre-trained network

(usually VGG-16 or VGG-19 network). The main difference from the pixel loss is that the computed loss is between the feature representations instead of the input and target images.

Despite the larger MSE of output images than other models have, the results are subjectively perceived as more sharp, and more details are recovered. However, the utilization of Sub-pixel layers in the last block (U-Net model) with the combination of perceptual loss produced the so-called checkerboard artifacts [40]. To reduce these artifacts there can be used a postprocessing filter. In this experiment, the Gaussian filter has been used with the kernel size $3 \times 3$. The output images after the filter look smoother than without using it and the MSE is also reduced.

U-Net+GEU2 architecture: As an attempt to avoid the checkerboard artifacts, the previous architecture was modified: after the input layer, the convolutional layer was used with the kernel size 7, instead of 3. The GEU block has a ReLU activation layer instead of the originally used sigmoid activation function. The next important thing is the utilization of upsampling layers in the U-Net model, instead of Sub-pixel layers. The number of feature maps starts from 64 instead of 32 and the bottleneck feature size is $4 \times 4$. As a result, the images are without checkerboard artifacts, but look not as sharp as in the previous architecture. The loss function used during the training is the feature reconstruction loss.

U-Net+GEU3 architecture: The third architecture of a neural network is a combination of GEU blocks and the second version U-Net. The main difference between them is the position of GEU blocks in the architecture. In this version, they are between the upscaling layers and the U-Net model. It allows us to extract more features from the upscaled feature maps. The U-Net has the same modified architecture as the U-Net+GEU2. The output images have some artifacts, but they are smoother than in its first version.

### 3.3. Training of Models

All methods (instead of the interpolation-based ones) were trained using the training samples to get their representative models. The performances of the models were evaluated by the test set. The methods were also compared to other interpolation methods. Single-frame methods obviously can only use one image input—frame number 4 was selected since it is the one most similar to the desired result. The multi-frame methods used all seven low-resolution frames as their inputs. NVIDIA GeForce GTX 1080 Ti and NVIDIA TITAN Xp cards were used for acceleration of time consuming computations and Tensorflow (https://www.tensorflow.org/) and PyTorch(https://pytorch.org/) frameworks were utilized to run training on GPU cards.

The implementations of single-frame super-resolution architectures were downloaded from public repositories (GitHub) and the methods were basically adapted to work with few modifications. These modifications included mainly path settings and re-designing for the scale factor 2. Some of the modifications could be done directly within the main file by parameter settings—the SRCNN and EDSR methods. For training the SRCNN method the following parameters were used: batch size: 128, number of epochs: 200, learning rate: 0.003. The EDSR method used 1000 iterations, Adam optimizer with a learning rate of 0.001 and batch size 10 for training. The SRGAN implementation had originally two Sub-pixel layers to obtain $4\times$ scaled images, but for the $2\times$ scaling it is enough to have just one layer. The training parameters are left without changes: the batch size is 8, the learning rate is 0.0001, the number of epochs for generator initialization is 100, for the adversarial learning of GAN the number of epochs is 2000. ESRGAN was also originally proposed for the scale factor 4, so for the scale factor 2 there these modifications were done: instead of two upsampling layers only one is used in the generator, the parameters in the linear layer of the discriminator are also changed to 2048 input features instead of 8192. The other parameters have been left without changes: batch size: 4, number of iterations: 40,000 and learning rate: 0.0001. The configuration of the training parameters of the single-frame super-resolution methods were the same as in their original proposals to assure the best performance.

We used a random search optimization method to select optimal parameters. As training is computationally and time-consuming, initial parameter values were inspired by other similar works. Results of each trained model with a different parameter configuration were subjectively evaluated. The number of epochs was estimated according to the results of the models to avoid over-fitting. The best training parameters were found as follows: number of steps per epoch 500, number of epochs 700, batch size 8, and learning rate 0.0001.

*3.4. Evaluation Metrics*

To objectively evaluate the quality of the resulting image, a couple of different metrics were used. The measurement of the quality of super-resolution images is not a straightforward task, especially since in the case of multi-frame methods there is not a single correct solution and in the case of face recognition, just a slight change of few pixels can have a significant impact on the overall perception of the face. At the same time, for reproduction purposes, these metrics should be based on easily accessible methods, preferably open-source methods, which should correspond to methods used in the past for this purpose in other papers. For this reason, we selected several different measures.

3.4.1. Objective Measures

The advantage of objective measures is that they provide a consistent metric for evaluation of the quality. All the metrics are described by mathematical equations or are based on open-source software, so the results can be reproduced. The following objective metrics were used.

Mean Square Error (MSE) measures the average of the squares of the difference between the actual value and estimated value, i.e., the difference between the images [41]. *MSE* is computed as

$$MSE = \frac{1}{m \cdot n} \sum_{i=1}^{m} \sum_{j=1}^{n} (I(i,j) - \hat{I}(i,j))^2, \tag{3}$$

where $I$ is the original image and $\hat{I}$ is the estimated image, both of width $m$ and height $n$. The best possible result of MSE can be 0—no difference. In general, the lower MSE value is considered better.

Peak signal-to-noise ratio (PSNR) is the ratio between the maximum possible power of a signal and the power of corrupting noise [42]:

$$PSNR = 10 \cdot \log_{10}(\frac{R^2}{MSE}), \tag{4}$$

where $R$ is the maximal variation in the input image data. The images are more similar to higher PSNR results (the unit of PSNR is a decibel).

Structural similarity (SSIM) [43] is a method for predicting the quality of digital television and images:

$$SSIM(I, \hat{I}) = \frac{(2\mu_I\mu_{\hat{I}} + C_1) + (2\sigma_{I\hat{I}} + C_2)}{(\mu_I^2 + \mu_{\hat{I}}^2 + C_1)(\sigma_I^2 + \sigma_{\hat{I}}^2 + C_2)}, \tag{5}$$

where $\mu$ is the mean, $\sigma^2$ is the variance, $\sigma$ is the covariance of $I$ and $\hat{I}$ and $C_1$ and $C_2$ are constants. The best possible result of SSIM is 1, a higher value represents better image similarity.

Cumulative probability of blur detection (*CPBD*) [44] is an index to evaluate the sharpness of an image:

$$CPBD = P(P_{BLUR} \le P_{JNB}) = \sum_{P_{BLUR}=0}^{P_{BLUR}=P_{JNB}} P(P_{BLUR}), \tag{6}$$

where

$$P_{BLUR} = P(e_i) = 1 - exp(-\left|\frac{w(e_i)}{w_{JNB}(e_i)}\right|^{\beta}), \tag{7}$$

where $w(e_i)$ is the width of the edge $e_i$, $w_{JNB}$ is the JNB (Just Noticeable Blur) [45] edge width which depends on the local contrast around the edge and $\beta$ is a parameter with a median value of 3.6 according to [46]. In the scope of this work, the sharpness is computed as the difference between the ground truth image CPBD and the predicted image CPBD.

Facial recognition similarity rate index (next FR rate), which is based on the Face-Recognition framework [31], is used as another objective metric. The project is not state-of-the-art in the area of face recognition, however, it is reproducible and accessible for free. The algorithm extracts 128 measures from the face based on a neural network called Facenet [32] and creates a feature vector. Similarity is computed from the feature vector of the ground truth image and the feature vector of the predicted image using the Euclidean distance. A lower value determines a better "face match", according to [32], distance under 0.6 is considered as the same person. Unfortunately, in some cases the Face-recognition algorithm failed to extract face features, therefore, it was not possible to measure the FR rate for the given image. The images which failed in feature extraction differ for each tested method. Because of that the FR rate (all) metric is computed from all the images in which it was possible to extract face features for the given tested model (no dependency on the other models). To make this metric more valuable, the FR rate ($\cap$) is defined as the metric computed only from the images in which all tested models were able to extract face features (intersection of the images over all the models).

The last metric—the face features extraction fail rate is simply defined as how many face images the Face-recognition failed at all (FR failed). This metric indicates the number of images in which it was not possible to extract face features, therefore, these images should not be used for face recognition. The best case for this metric is 0 (which is the minimum), the worst one is the total size of the test set—in our case 2500 (which is the maximum).

3.4.2. Subjective Human Evaluation

During the experimenting of the various approaches, we observed that the proposed methods (i.e., U-Net+GEU) seem to be promising when looking at the resulting reconstructed image as was shown in Figure 2. Unfortunately, this method does not have such a significantly better result when using objective metrics evaluation. Because the subjective human evaluation has an important role during results evaluation, especially for possible real application cases, evaluation based only on objective metrics would displace the best method for human treating by some other methods with promising objective metric results, but useless for humans.

The questionnaire data set contains 126 images and tries to cover as many various images (lighting, people, angles, ...) as possible. This set was used to measure the subjective human evaluation in the form of a questionnaire. The questionnaire user interface contains 126 sections with predictions for each method (12 images in total, input, and ground truth excluded). The names of the methods were not shown to get fair (correct) results.

One hundred volunteers filled out the questionnaire with the objective: "Please, mark the image that you think is the best for face recognition purposes, for example—robbers identification by police".

Simply said, select the best image in your opinion. To get the most diverse results, the questionnaire was sent to people from all around the world, including both genders, different ages, and people from various work environments. Volunteers were selected randomly including not only students and graduates of technical universities, but also people without any technical background employed in a variety of fields. A few of the selected volunteers were also experts involved in the image processing and face recognition area.

## 4. Results

The most successful approaches of recent years were based mostly on approaches based on neural networks, and this paper primarily builds upon those methods too. We have examined in total eight different architectures of neural networks: four state-of-the-art single-frame methods. Specifically, they were EDSR, SRCNN, SRGAN, and ESRGAN. Furthermore, we introduced

four multi-frame methods, and we called them U-Net+ResBlock, U-Net+GEU, U-Net+GEU2, and U-Net+GEU3. Some of them take inspiration from the most successful single-frame based methods and are modified to work with multiple frames. These architectures were pre-selected after experimentation with many different combinations of architectures and their blocks, where those four have been shown to be the most promising. For completeness, several generally known interpolation methods were also included in the experiment for the purpose of comparison with the earlier mentioned methods. Gaussian filter was applied to the output of U-Net+GEU to suppress checkerboard artifacts [40] and due to an interesting results comparison, it is considered as an individual method. Thus, a total of 12 different super-resolution methods were evaluated and compared. Although GAN networks have been very successful in this area in recent years, the proposed multi-frame architectures were not included in the experiment. The reason for this exclusion was that it was shown they are too creative as was discussed in Section 2.2. Although the faces look right, they cannot be used for person identification since they generate completely fake faces, which are not primarily based on their input information. In Section 5) the results are discussed.

*4.1. Resulting Dataset*

Based on the process described in Section 3.1.2, we created an MLFDB dataset that contains a total of 17,426 samples. Each sample contains a sequence of 7 low-resolution face images ($32 \times 32$ px). Each of these sequences has its corresponding label in the $64 \times 64$ px resolution. The label was created from the middle image of the sequence (frame number 4). When it was possible, labels in the higher resolution were also provided. Therefore, some of the sequences also have labels in $128 \times 128$ px (12,165 samples of the whole dataset, i.e., 70%) and $256 \times 256$ px (4199 samples, i.e., 24%) resolution. An example of dataset samples is shown in Figure 9.

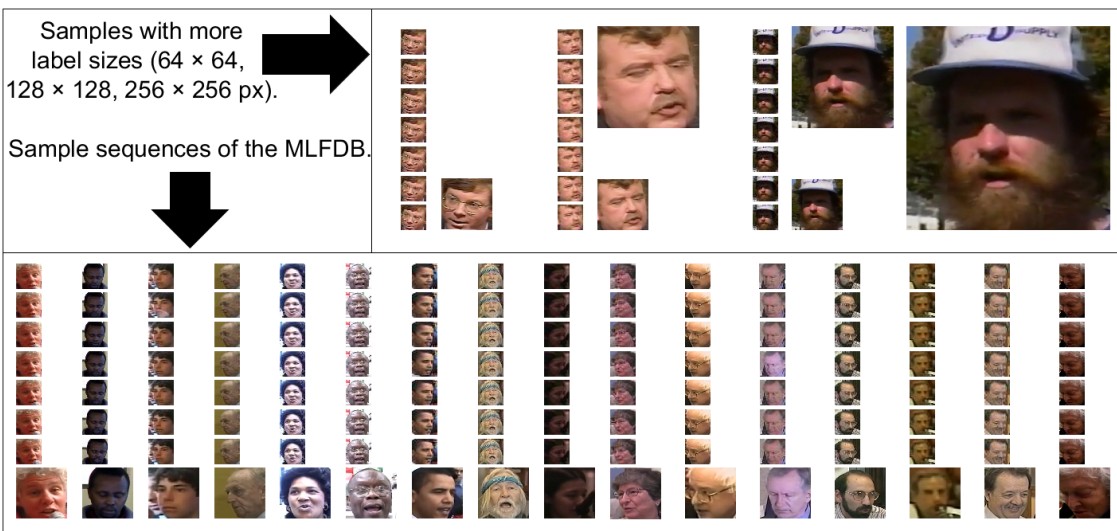

**Figure 9.** An example of sequence samples of the dataset.

This dataset of 17,426 samples was divided into a training set (12,200 samples, 70%), test set (2600 samples, 15%), a private test set (2500 samples, 15%), and a test set for a questionnaire (126 samples) for subjective human evaluation. The training dataset is intended for the training and optimization of super-resolution methods. The test set is only for the evaluation of results. This dataset should not be used for optimization or search of parameters to avoid over-fitting. The private test set is kept private at the Brno University of Technology and will be used for evaluation on request to ensure the objectivity of the results (http://splab.cz/mlfdb/#results). The questionnaire data set is intended for evaluation by human reviewers. This is mainly because results measured by objective mathematical methods do not always correspond with the quality perceived by human evaluators.

*4.2. Methods Comparison*

All objective measures described in Section 3.4.1 were used for the performance evaluation of the models: SSIM, MSE, PSNR, sharpness as the difference between the ground truth image and the predicted image, face features extraction fail rate (FR failed) and face recognition similarity rate index, i.e., FR rate. Please note that FR rate (all) is computed only from the images where the face features extraction was successful for the given resulting method. FR rate ($\cap$) metric is computed from the intersection of the images with successful face features extraction by all methods. The amount of these images is 1518 of all 2500 test images.

The results of all 12 methods are placed all together into Table 2 for the compendious comparability of results between the single-frame and multi-frame approaches.

**Table 2.** Single-frame and multi-frame methods performance comparison—mean.

| Method | SSIM | MSE | PSNR | Sharpness | FR Rate (All) | FR Rate ($\cap$) | FR Failed |
|---|---|---|---|---|---|---|---|
| bicubic | 0.806 | 227.196 | 25.654 | 0.084 | 0.476 | 0.473 | 674/2500 |
| bilinear | 0.809 | 217.368 | 25.769 | 0.158 | 0.471 | 0.468 | 693/2500 |
| lanczos | 0.802 | 238.247 | 25.504 | 0.024 | 0.483 | 0.480 | 700/2500 |
| EDSR | 0.780 | 286.999 | 24.687 | −0.025 | 0.495 | 0.494 | 701/2500 |
| SRCNN | 0.815 | 216.940 | 25.771 | 0.209 | 0.461 | 0.456 | 643/2500 |
| SRGAN | 0.760 | 320.981 | 24.234 | −0.078 | 0.511 | 0.510 | 771/2500 |
| ESRGAN | 0.788 | 281.192 | 24.604 | −0.016 | 0.488 | 0.485 | 470/2500 |
| U-Net+GEU | 0.760 | 276.459 | 24.383 | −0.129 | 0.507 | 0.503 | 317/2500 |
| U-Net+GEU+filter | 0.815 | 234.552 | 25.196 | 0.239 | 0.464 | 0.458 | 394/2500 |
| U-Net+GEU2 | 0.799 | 256.511 | 24.878 | 0.031 | 0.472 | 0.467 | 259/2500 |
| U-Net+GEU3 | 0.799 | 243.433 | 25.135 | 0.004 | 0.471 | 0.466 | 270/2500 |
| U-Net+ResBlock | 0.812 | 239.114 | 25.141 | 0.130 | 0.468 | 0.463 | 581/2500 |

As can be seen from the results, the SRCNN method can be considered as the most suitable method in the way of the highest number of best objective metrics (the best results for 5 from 7 metrics), while not considering subjective human evaluation. However, the sharpness and the FR failed metrics of the SRCNN method are significantly worse when comparing the results of other methods.

The best face recognition extraction fail rate (FR failed) has the U-Net+GEU2 method that failed only in 259 cases out of 2500 compared to the SRCNN method that failed in 643 cases out of 2500. Moreover, the sharpness difference between these two methods is noticeable. It should be noted that the results described above only presents the performance of methods according to the number of the best objective measures and it does not take into account their significance (i.e., SSIM and FR failed are probably more valuable than MSE). The important metric—subjective human evaluation—is not included in these results, therefore, they should not be generally taken as the final outcomes.

U-Net+GEU2 significantly outperformed other methods by the subjective human evaluation (see results in Section 4.3), but among the objective metrics it had only one best result—face features extraction fail rate (FR failed). The other values are not so good comparing to the SRCNN method, but the differences between them are significant unlike for the face features extraction fail rate metric where the difference is evident. Because of this, and due to the different importance of each metric, these values and their comparison are better readable by a normalized representation into the range (0–1) (see Table 3). Minimum and maximum values were determined from all method's values for the given metric. In the case of the observed metric, where the higher value is better (SSIM, PSNR, subjective evaluation), the normalized value is computed as $1 - norm(x)$ to have all values with the same logic—lower value is better (it is considered as a "penalization"). The final score is then simply the sum of all normalized values, sum1 includes only the objective metrics and sum2 also includes the subjective human evaluation—sub. ev.

**Table 3.** Normalized values comparison of all methods to get the final score.

| Method | SSIM | MSE | PSNR | Sharp. | FR Rate (∩) | FR Failed | Sub. ev. | Sum1 | Sum2 |
|---|---|---|---|---|---|---|---|---|---|
| bicubic | 0.164 | 0.099 | 0.076 | 0.340 | 0.315 | 0.811 | 1 | 1.805 | 2.805 |
| bilinear | 0.109 | 0.004 | 0.001 | 0.655 | 0.222 | 0.848 | 0.989 | 1.839 | 2.828 |
| lanczos | 0.236 | 0.205 | 0.174 | 0.085 | 0.444 | 0.861 | 0.999 | 2.005 | 3.004 |
| EDSR | 0.636 | 0.673 | 0.705 | 0.703 | 0.089 | 0.863 | 0.977 | 3.669 | 4.646 |
| SRCNN | 0 | 0 | 0 | 0.872 | 0 | 0.750 | 0.978 | 1.622 | 2.600 |
| SRGAN | 1 | 1 | 1 | 1 | 0.315 | 1 | 0.995 | 5.315 | 6.310 |
| ESRGAN | 0.491 | 0.618 | 0.759 | 0.051 | 0.537 | 0.412 | 0.844 | 2.868 | 3.712 |
| U-Net+GEU | 0 | 0.572 | 0.903 | 0.532 | 0.870 | 0.113 | 0.570 | 2.990 | 3.560 |
| U-Net+GEU+filter | 0 | 0.169 | 0.374 | 1 | 0.037 | 0.246 | 0.578 | 1.844 | 2.422 |
| U-Net+GEU2 | 0.281 | 0.380 | 0.581 | 0.115 | 0.204 | 0 | 0 | 1.571 | 1.571 |
| U-Net+GEU3 | 0.281 | 0.255 | 0.414 | 0 | 0.185 | 0.021 | 0.204 | 1.166 | 1.370 |
| U-Net+ResBlock | 0.055 | 0.213 | 0.410 | 0.536 | 0.130 | 0.629 | 0.834 | 1.973 | 2.807 |

As can be seen from Table 3, after a desirable representation of the results and their overall score in form of the sum, U-Net+GEU2 and U-Net+GEU3 methods have a better overall score than the SRCNN method even without using the subjective human evaluation metric (sum1). The overall score including the subjective human evaluation metric (sum2) enlarges the difference between the methods almost twice as much. The U-Net+GEU+filter method has a worse overall score (sum1) than the SRCNN method (1.844 vs. 1.622) but the results are opposite (2.422 vs. 2.600) after including the subjective human evaluation (sum2).

An example of resulting images (predictions) on unseen data from all 12 methods is shown in Figure 10. Each row represents one input sequence and each column shows resulting images for each method. Finally, there is a ground truth image. The presented samples are not part of the MLFDB test set because the labels (ground truth images) should not be publicly available, therefore, these sequences were created for presentation purposes only.

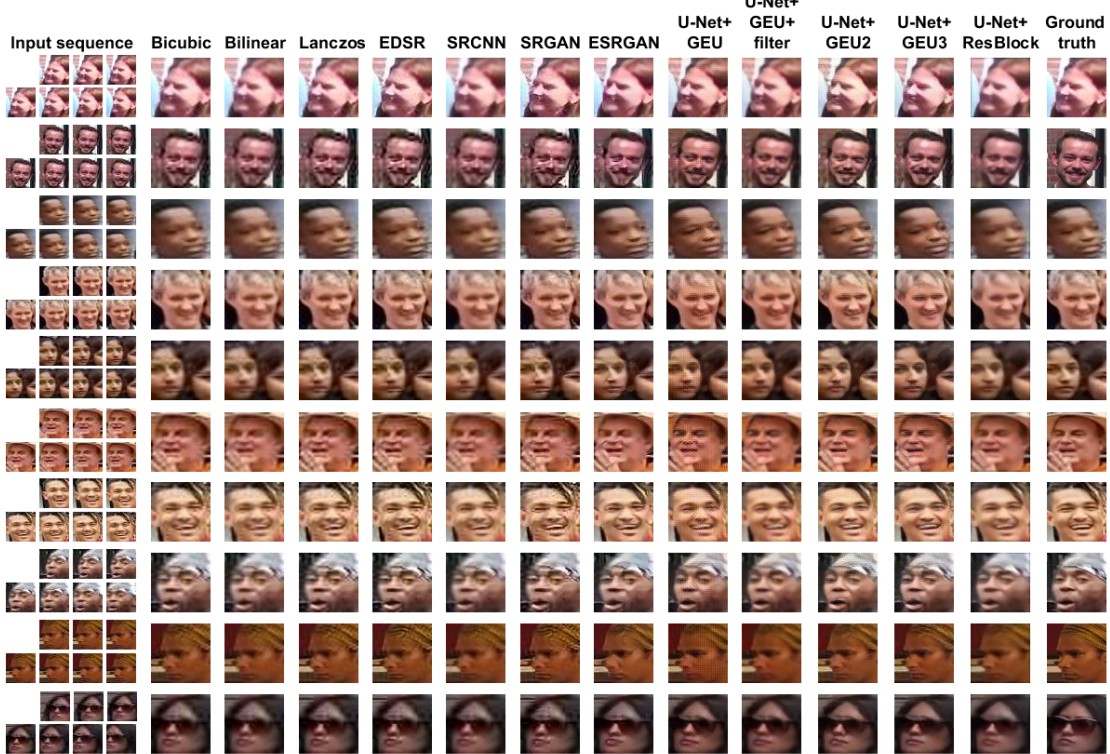

**Figure 10.** Examples of the resulting methods. All seven 32 × 32 px images of the input sequence were used for the multi-frame methods and the 4th image for single-frame approaches was used.

As expected, the interpolation-based methods cannot reconstruct the image in a better quality, but the differences between them are noticeable. Results of the EDSR and SRGAN methods look similar to the interpolation-based methods. The SRCNN method, which has the best results among objective metrics, predicts the resulting image a little better, compared to other single-frame methods, unlike the ESRGAN whose prediction quality is much better (sharper), but it sometimes creates visible deformation artifacts. All multi-frame methods provide notably better results in contrast to single-frame methods. The U-Net+ResBlock method creates a bit smooth blurry images, which is annoying for human perception. The U-Net+GEU+filter method creates similar results, but they look not so remarkable. The resulting images of other U-Net+GEU based methods act similarly to each other but are significantly better compared to all other methods.

### 4.3. Questionnaire Results

The subjective human evaluation metric was examined by a questionnaire in which 100 people had to select the best image in their opinion, among 126 comparisons of 12 methods. In total 12,600 "votes" were available and they were distributed to all methods by individual preferences of attending people. The visualization of subjective human evaluation results is illustrated in Figure 11.

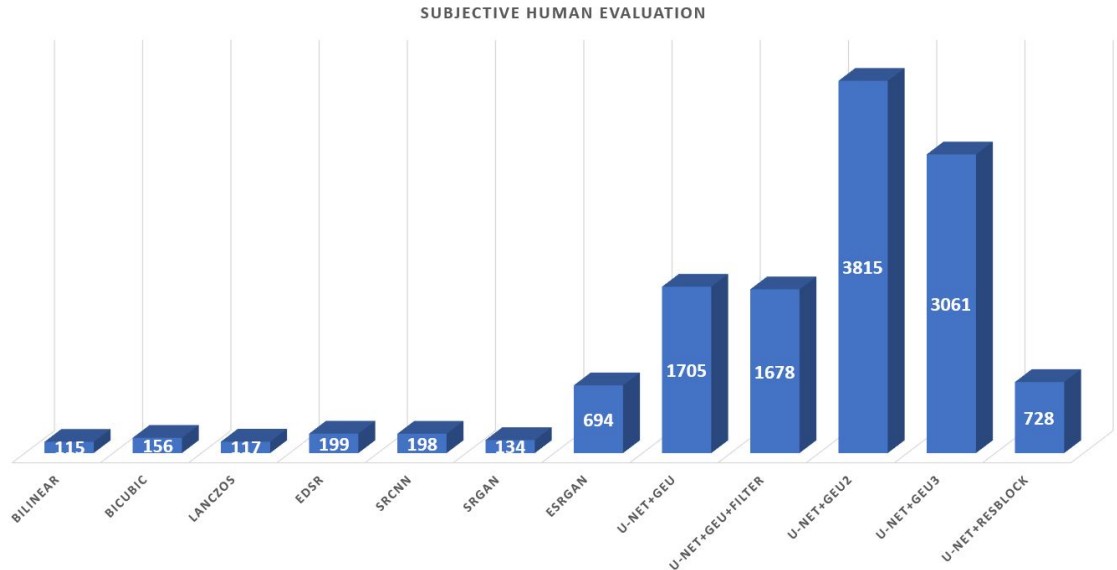

**Figure 11.** Results of the questionnaire—subjective human evaluation. 100 people attending filled out the questionnaire containing 126 comparisons of 12 methods.

From the first look at the chart, it is clear that all U-Net+GEU methods significantly overcome other methods in human perception. 3815 votes from 12,600 belong to the U-Net+GEU2 method. U-Net+ResBlock and ESRGAN with approximately 700 votes are following the U-Net+GEU base methods. ESRGAN significantly exceeds all single-frame methods and as only one single-frame method can be compared to results achieved by the multi-frame method—U-Net+ResBlock, which is the worst among multi-frame methods, by human perception. Remarkably, that SRCNN has the best objective metrics, but ESRGAN is much more successful by human perception. A similar situation is the same for multi-frame methods where the U-Net+GEU+filter has better objective metrics, but the U-Net+GEU2 has a significantly better subjective human evaluation.

## 5. Discussion

### 5.1. Dataset

Unfortunately, it is quite challenging to automatically count unique faces in this dataset. Due to the high number of sequences, it is almost impossible to remember all the faces and the manual approach is, therefore, not possible as well. For this reason, we used an automated approach which is based on the Face-recognition framework and can measure a similarity of faces. This framework returns zero value when two faces are of the perfect similarity. A higher value than zero means less similarity of these faces, which might be caused not just by different faces, but also because faces were taken from different angles, have lower quality, etc. According to the project authors, the default threshold value by which the similarity value of two faces should express the same person is ≤0.6. This value is recommended for facial recognition purposes (computed on the LFW dataset). However, as can be seen in Table 4, a threshold value of 0.6 is improper for this purpose. The reason for this is that MLFDB contains face images that are on the borderline when they start being useful for the identification (i.e., lower quality than the LFW dataset). For the threshold value of 0.6, only 76 sequences were considered unique, which is after manual review and definitely not the truth. Based on the empirical experimentation with different values, we found the threshold value 0.45, which shows good performance. These results were then validated manually on a selected subset.

**Table 4.** Different people in the dataset using the Face-recognition framework similarity threshold.

| Threshold | 0.6 | 0.5 | 0.45 | 0.4 | 0.35 |
|---|---|---|---|---|---|
| different people | 76 | 2378 | 6651 | 10,822 | 13,369 |

Sequences from one video source were also filtered using the SSIM metric, where we used the threshold value 0.7. Only the ground truth images were used for this filtering. This filter ensures that sequences even of the same person are kept when there is a different scale, lighting, angles, etc. (it was discussed in Section 3.1.2).

The dataset is further divided into public and private parts. The reason for this is to increase the objectivity of performance evaluation. In the case all the parts would be published, they can still be used for the training. Such results would with high probability outperform the results of other teams, but those results would be over-fitted and would absent generalization of course. Because of that, the test part of the dataset was split into two parts: the test-public and test-private. The test-private was also released but without ground truth images. Evaluation of this private dataset is possible on the MLFDB website by submitting achieved results. The evaluation requests are limited by time to avoid already mentioned over-fitting.

### 5.2. Questionnaire

The problem of face image super-resolution is significantly more challenging than general super-resolution tasks. It is clear that in every super-resolution method, some particular errors and mistakes must be created since the input image contains less information than there is expected in the resulting image. Thus, some percentages of the pixels are estimated. In the case of general super-resolution methods, a mistake in the image is often hard to be noticed and can be ignored. However, when the image is checked for some details, typically texts, brand logos, or other well-known complex shapes, some errors are apparent at first glance. This is also the case for facial images. For example, if the resulting image of a face is missing an eye, or it has three eyes, it might be for metrics like PSNR or MSE neglectable error, but for human perception, it can be very disruptive. The same situation is with two noses or no nose, the asymmetrical appearance of the eyes or otherwise deformed face.

Another topic that should be taken into consideration is that the main objective is not just to generate images that are good looking. A more important criterion is that the resulting facial image should be of such a quality that the identification must be possible. The studied images are at the borderline where images start being useful for the recognition. When considering each particular low-resolution image from a sequence alone, it is not possible or it is difficult to recognize a person. Nevertheless, the resulting higher resolution image should be of sufficient quality, in which it can be used for this purpose. For this reason, we addressed several volunteers and asked them to fill out the questionnaire. The purpose of the questionnaire was to obtain feedback regarding the quality of the achieved face images from the human perspective. The volunteers saw all the results of the super-resolution methods, and they were asked to mark the result, which they consider is the best.

The U-Net based methods were their main choices, and the interpolation-based methods were almost not selected. An interesting finding was that the SRGAN method was considered as the worst one. The decisions were in general made based on the essential face features clearness (same eyes, nose, mouth)—the entirety of the face. However, some cases degraded teeth and face borders so much that the preferred image was with a lower quality image but with an entire face. The ESRGAN method was the only method that had significant deformations of the face. Another interesting information was that people intuitively prefer blurred edges instead of sharp edges and no artifacts, even when the image looks good. These feedbacks correlate with the objective results, except for the ESRGAN method. It is the second sharpest method and it has got the most votes among all single-frame methods from the questionnaire.

### 5.3. Results Comparison

Results were evaluated by two approaches, using objective mathematical metrics (see Table 2) and subjective evaluation based on the human volunteers' votes (see Figure 11). Furthermore, since every objective metric evaluates the images using a different point of view, we have merged all the results (using their normalized representation) and show the overall score which is denoted as sum1 and sum2 (see Table 3).

According to the subjective metrics, the multi-frame methods achieved the best results. Furthermore, after the normalization of the objective metrics values and improved representation (instead of absolute values) they have had even better than the SRCNN method, which was evaluated as the best as pointed in Table 2. The probable reason is that the results of individual metrics of multi-frame methods are very close to the best values among the tested methods (basically the SRCNN method) and primarily they do not have any significant outliers among all metrics compared to single-frame methods, especially the SRCNN method.

No GAN-based multi-frame super-resolution methods were used in this work. The reason for this is their generative nature, which was discussed in more detail in Section 2.2. In the scope of this work, there were conducted experiments with GANs, but these experiments concluded, the current methods should not be used for person identification purposes. It should be emphasized that general super-resolution and super-resolution for identification purposes have different objectives.

Nowadays, one of the obstacles to the faster development of these methods is the lack of objective metrics which will better correspond to human perception. In this paper, we introduce an extended set of metrics like SSIM, PSNR, etc. Unfortunately, it was not a rare case when the best results selected by the human volunteers did not match the top results by the objective metrics. Because of that, the deep learning models created images with small checkerboard artifacts such as small different colour tints or small face re-positioning, which affect these metrics. Those issues become even more important if there are no ground-truth images available, i.e., the model is used in real situations. On the other hand, it is probably better to mark several suspicious face images with lower accuracy, rather than not selecting any of the faces. For example, it is better to preemptively check ten people instead of no-one when looking for dangerous suspects.

*5.4. Face Recognition Based Metrics*

For all 2500 test images, the face recognition similarity rate index (FR rate (all)) was computed. The same metric was also computed for the subset of the test dataset where face recognition was successful (1518 samples). Only those samples are considered to be successful, where all the models created such a resulting image, which succeeded with face recognition. There was an assumption that the methods with better face features extraction fail rate (lower FR failed) will have a better face recognition similarity rate index on those subset images instead of the whole test set. However, according to achieved results, it seems that this has only a minimal effect.

A more interesting metric is the face features extraction fail rate (FR failed), i.e., in how many cases it was not possible to extract successfully face features. According to the results, the U-Net+GEU2 method has the lowest number of cases where face recognition failed. It was also examined in how many cases it failed on the same images. The number of different images that are not in the set of failed images of the U-Net+GEU2 method is used to make the output more interpretable. These values are shown as the first value in Table 5. The second value after the slash represents the overall difference from the U-Net+GEU2 FR failed metric.

**Table 5.** Face recognition (FR) failed differences compared to the U-Net+GEU2 method.

| Method | Differences |
|---|---|
| bicubic | 12/415 |
| bilinear | 9/434 |
| lanczos | 16/441 |
| EDSR | 18/442 |
| SRCNN | 11/384 |
| SRGAN | 15/512 |
| ESRGAN | 35/211 |
| U-Net+GEU | 38/58 |
| U-Net+GEU+filter | 16/135 |
| U-Net+GEU3 | 55/11 |
| U-Net+ResBlock | 10/322 |

The best match of failed images is paradoxically for the bilinear interpolation. On the other hand, the bilinear interpolation failed in 693 cases instead of the U-Net+GEU2 method that failed in 259 cases (the difference is 434) and the bilinear interpolation set of failed images is big enough to match the majority of the U-Net+GEU2 set of images. An interesting finding is an opposite issue. The U-Net+GEU3 method has almost the same number of FR failed metric (the difference is only 11 cases), but its set of images differs in 55 cases compared to the U-Net+GEU2 method.

After further examination of the method, there were no patterns found (the same interpolations, compression, images background, etc.), which would explain why there are so different results. However, there are some diversities between U-Net+GEU2 and U-Net+GEU3 methods in some cases. Figure 12a shows a detailed view of the resulting quality created using U-Net+GEU2 and U-Net+GEU3. Surprisingly, there are also contrary cases, but they are not so significant as illustrated in Figure 12b. U-Net+GEU3 method failed in face features extraction in both cases. There are probably two essential explanations. First, human perception differs from the machine learning image processing and how the quality of the input image is assessed. Second, the well-known and used Face-recognition framework does not work as well as presented, at least for this case. Among the images which failed in face features extraction are a couple of profile images (see examples of labels in Figure 12c). The face features extraction works in this case, but with what accuracy? How do face landmark estimations and face transformations work in these cases?

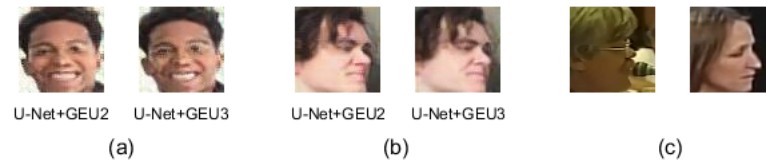

U-Net+GEU2　　U-Net+GEU3　　　　　U-Net+GEU2　　U-Net+GEU3

　　　　　(a)　　　　　　　　　　　　　　(b)　　　　　　　　　　　　　(c)

**Figure 12.** Examples of low face feature extraction performance. (**a**,**b**) U-Net + GEU2 (passed) and U-Net + GEU3 (failed) methods comparison. (**c**) Examples of profile face images of database samples that passed face features extraction as well during the database creation filtration step.

### 5.5. Benchmark and Leaderboard

MLFDB dataset is the first multi-frame facial dataset of comparable size, which has the potential to support the development of this field. Another important step is to set a baseline in the form of results from various methods, so anyone can compare the achieved results with new methods.

The general problem of many datasets is that they can be easily over-fitted. This often happens when parameters are optimized not just using the training set, but also the test set. In the situation when everything is publicly available, it is not possible to have control over this. Millions of trails can lead in some cases to results that do not reflect the real performance of the method. Sometimes there is not clearly defined how the comparison metrics should be computed, so the progress is difficult to be objectively evaluated.

To deal with this problem, we decided to create an automatic benchmark (inspired by systems like Kaggle (https://www.kaggle.com/)) for an objective results comparison on this dataset. Ground truth images of the private part of the test dataset are not publicly available, and thus metrics are computed in the same manner as they were computed in this paper. For this purpose, the MLFDB server is available online, where anyone can submit their results (http://splab.cz/mlfdb/). The final score will be computed as sum1 described in Section 4.2. The number of results of computation queries will be limited according to the rules of the benchmark.

### 5.6. Future Research

There are plenty of ways how to follow up on this work. Especially, it could be interesting to experiment with models with a higher scale factor, for example, $4\times$ or $8\times$, as was pointed out in the example in Figure 2. The MLFDB dataset has all ground truth images in resolution $64 \times 64$ px, but there are also many images in even higher resolution—$128 \times 128$ px (12,165 samples) or $256 \times 256$ px (4199 samples). Another way worth trying could be changing the number of images in sequences or a more complex and better utilization in real situations would be an application of some feature selection [47]. The feature selection in general leads to a better performance because of focusing on the important parts of input data and removing outliers or noisy data that cause model inaccuracy.

### 6. Conclusions

Forensically trained facial reviewers are still considered to be one of the most accurate approaches to person identification, especially in the case of low-resolution or low-quality videos. The human brain can utilize information not just from a single image but also from a sequence of faces (i.e., videos) and, even in the case of low-quality records or a long distance from a camera. They can accurately identify a person. For computer methods, this remains a challenge. However, on the other hand, they have the potential to support human reviewers and help to pre-process the data and extract as much information from the data as possible. One of such a use-case would be, for example, to reconstruct the facial image in higher quality from a video, which can police use for search and announce it in newspapers.

This paper introduced a large-scale face dataset containing 17,426 sequences of face images. The dataset covers different races, ages, lighting conditions, and many types of different camera devices. Records were obtained from the real environment, including common defects and imperfections that occur in the videos (e.g., compression artifacts, blur, etc.) According to our knowledge, it is the first

dataset of a comparable size containing sequences of face images. The paper also introduces a new multi-frame face super-resolution method. This method has been proven to produce better results than single-frame methods which are considered as state-of-the-art, and they are one of the most effective and often compared methods in this research area. Therefore, the hypothesis that the multi-frame method produces better results than the deep learning-based single-frame methods was proven.

The results were also evaluated using several objective metrics and also subjectively by volunteers where the U-Net+GEU2 method has the best results regarding human perception, and the U-Net+GEU3 method achieved the best overall score for objective metrics. The proposed method is not generative and it was proven to improve the quality of the face image. The source code and the dataset were released, and the experiment is fully reproducible.

The new and unique MLFDB dataset has been published for studying face super-resolution from sequences of images (i.e., multi-frame problem). The benchmark with given rules and metrics was created, and results achieved in this paper were used as an initial starting point and as a challenge to other researchers. They are presented in the leaderboard table as a part of the benchmark. Regarding results themselves, it was found that human perception plays a significant role during model evaluation. Therefore, it is essential to use some subjective human evaluation, for example, in the form of questionnaires or surveys.

**Author Contributions:** Conceptualization, M.R. and R.B.; methodology, A.M.; software, M.R. and A.M.; validation, M.R. and R.B.; formal analysis, A.M. and M.R.; investigation, A.M.; resources, R.B.; data curation, M.R. and A.M.; writing—original draft preparation, M.R.; writing—review and editing, R.B.; visualization, M.R. and A.M.; supervision, R.B. All authors have read and agreed to the published version of the manuscript.

**Funding:** This research was funded by the Interreg Central Europe niCE-life by the grant CE1581.

**Acknowledgments:** Research described in this paper was financed by the Interreg Central Europe niCE-life by the grant CE1581. We would like to thank all of the volunteers that attended the questionnaire, especially the ones who provided the feedback.

**Conflicts of Interest:** The authors declare no conflict of interest.

## Abbreviations

The following abbreviations are used in this manuscript:

| | |
|---|---|
| CelebA | Large-scale CelebFaces Attributes |
| CCTV | Closed circuit television |
| CNN | Convolutional Neural Network |
| CPBD | Cumulative Probability of Blur Detection |
| CVPR | Conference on Computer Vision and Pattern Recognition |
| DB | DataBase |
| DeepSUM | Deep neural network for Super-resolution of Unregistered Multitemporal images |
| EDSR | Enhanced Deep Super-Resolution Network |
| ESRGAN | Enhanced Super-Resolution Generative Adversarial Network |
| EDVR | Enhanced Deformable Convolutional Network |
| FERET | Face Recognition Technology |
| FR | Face Recognition |
| FRVSR | Frame-recurrent video super-resolution |
| FSRGAN | Face Super-Resolution Generative Adversarial Network |
| FSRNet | Face Super-Resolution Network |
| GAN | Generative Adversarial Network |
| GPU | Graphics Processing Unit |
| JNB | Just Noticeable Blur |
| LFW | Labeled Faces in the Wild |
| MLFDB | Multi-frame Labeled Faces Database |
| MSE | Mean Square Error |
| NN | Neural Network |
| PCD | Pyramid, Cascading and Deformable convolutions |

| PSNR | Peak signal-to-noise ratio |
| --- | --- |
| PubFig | Public Figures Face Database |
| PULSE | Self-Supervised Photo Upsampling via Latent Space Exploration of Generative Models |
| ReLU | Rectified Linear Unit |
| ResNet | Residual Network |
| ResBlock | Residual Block |
| SR | Super-Resolution |
| SRCNN | Super-Resolution Convolutional Neural Network |
| SRGAN | Super-Resolution Generative Adversarial Network |
| SSIM | Structural similarity |
| TSA | Temporal and Spatial Attention |
| UR-DGN | Ultra-resolution by discriminative generative network |
| UVT | Unpaired Video Translation |
| VSR | Video Super-Resolution |
| YOLO | You Only Look Once |

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
