# Peer review of "Multi-Frame Labeled Faces Database: Towards Face Super-Resolution from Realistic Video Sequences"

_applsci, doi:10.3390/app10207213_

Round 1

Reviewer 1 Report

This paper introduce a new method for face super-resolution and a large face dataset containing sequences of images, which covers various races, ages, lightening conditions and camera device types. It also introduces a new multi-frame face super-resolution method and compares this method to the state-of-the-art methodologies. The paper is generally well written, but I do have some concerns in several points here below detailed:

lines 52-73: The authors enumerate several contributions of the paper but in a rather sparse order, in a way that turns difficult to fully understand it. I think the paper would benefit from listed items which highlight better each novelty/improvement.

Section 2: after reading this section I would like to find a table summarizing all the aspects that this work improves compared to state-of-the-art solutions.

Section 3.3: the authors should detail how they come to those parameters for the training. It is not clear which method they used for obtaining the parameters used for each Neural Network configuration. 

line 451: "100 volunteers attended the questionnaire with the objective:" Do the authors assessed the background/expertise of such volunteers? It is worth to show the volunteers background/expertise.

lines 479-484: authors use the pattern metric (how to read it) I suggest to move the measures interpretation (with the values range etc.) in the previous section, i.e. 3.4.1. Objective measures

Sections 4.2 and 4.3, and more specifically Tables 1,2,3,4: the authors show here the results of the various methods experimented and conclude that some methods (e.g., Unet+GEU+filter) are better than others. Please, add a statistical test assessing which one is better and the significance.

I wish to compliment to the authors for the fantastic figures on the paper!

Reviewer 2 Report

This manuscript introduces a new face-recognition database used for multi-frame super-resolution methods. The database is well-designed in term of variety.
1. However, the proposed method uses U-Net framework without decent contribution.
2. The authors claim that "the proposed method outperforms the current state-of-the-art" but it is not proved by the experimental results.
3. Why do the authors compare single-frame methods while they use multi-frame methods?
4. The manuscript is not well written and not clear. The structure/outline should also be revised.
5. In subsection 3.4.1, the authors must cite objective quality measurements.

Round 2

Reviewer 2 Report

  1. First paragraph of section 3 should be moved to section 4. Also in this paragraph, something is incorrect in this sentence: "We have examined in total 9 different architectures of neural networks: four state-of-the-art single-frame methods were selected for the experiment and furthermore we introduced four multi-frame methods, where some of them are modified former approaches". Please explicitly specify the examined methods.
  2. 'y' is used in equations (1) and (2) with different meanings. Please revise this.
  3. The notation for images should be in capital form (equations (2) and (5)).
  4. No explaination for 'n' and 's' in Figures 6, 7, 8.
  5. Line 384-386, why does Gaussian filter make the images more sharp? Please explain because I think it is technically incorrect.
  6. Move subsubsection 3.2.1 to another section. Sub-section 3.4 should be moved to section 4.
  7. In section 4, why do the authors use both MSE and PSNR for comparison since they are related?
  8. Sub-section 4.4 (subjective metrics results) lies between other sub-sections which cover objective metrics results. Section 4 's outline should be rearranged.
  9. Why do the authors write sub-sections 4.2 and 4.3 to separately compare single-frame and multi-frame approaches, respectively, while sub-section 4.5 summarizes and combines them all? Table 6 is essentially the combination of Table 2 and Table 4. I think the authors should remove sub-sections 4.2 and 4.3.
  10. The visual quality of Figure 11 is low. It should look as clear as Figure 2 so the readers can see how good is the proposed approaches.
  11. Line 733: please consider using 'outperform'. As I mentioned in the previous review, it is not correct to say "the proposed methods outperform the current state-of-the-art ones."
  12. Consistently use artefact or artifact.
  13. I recommend the authors using an English editing service. I don't see any improvement in English writing in comparison with the previous version.

Round 3

Reviewer 2 Report

The authors resolved all of my comments. I think the manuscript is now adequately good to be published in the journal.